# Impact of sleep duration on executive function and brain structure

Xin You Tai [1,2✉], Cheng Chen[3], Sanjay Manohar [1,2,3] & Masud Husain [1,2,3]

Sleep is essential for life, including daily cognitive processes, yet the amount of sleep required for optimal brain health as we grow older is unclear. Poor memory and increased risk of dementia is associated with the extremes of sleep quantity and disruption of other sleep characteristics. We examined sleep and cognitive data from the UK Biobank ($N = 479,420$) in middle-to-late life healthy individuals (age 38–73 years) and the relationship with brain structure in a sub-group ($N = 37,553$). Seven hours of sleep per day was associated with the highest cognitive performance which decreased for every hour below and above this sleep duration. This quadratic relationship remained present in older individuals (>60 years, $N = 212,006$). Individuals who sleep between six-to-eight hours had significantly greater grey matter volume in 46 of 139 different brain regions including the orbitofrontal cortex, hippocampi, precentral gyrus, right frontal pole and cerebellar subfields. Several brain regions showed a quadratic relationship between sleep duration and volume while other regions were smaller only in individuals who slept longer. These findings highlight the important relationship between the modifiable lifestyle factor of sleep duration and cognition as well as a widespread association between sleep and structural brain health.

[1] Nuffield Department of Clinical Neuroscience, University of Oxford, Oxford, UK. [2] Division of Clinical Neurology, John Radcliffe Hospital, Oxford University Hospitals Trust, Oxford, UK. [3] Department of Experimental Psychology, University of Oxford, Oxford, UK. ✉email: xin.tai@ndcn.ox.ac.uk

Sleep is essential for day-to-day life and plays a role across a range of physiological functions[1]. The impact of sleep on the brain is particularly substantial and better understanding of this may play a key role in maintaining healthy cognitive ageing as people grow older. New insights have emerged in recent years around sleep and cognitive processes such as memory consolidation and learning[2], as well as clearance of neurodegenerative proteins from the brain related to development of Alzheimer's disease[3,4]. Crucial questions nevertheless still remain about sleep and brain health. Firstly, what is the optimal duration of sleep for cognitive functioning and does this change across the lifespan? And secondly, how does sleep relate to brain structural health?

Observational studies have offered some insights into these questions. For example, extremes of overnight sleep duration are related to worse cognition in middle-to-late life individuals. Findings have associated sleeping too short[5–7], too long[8–10] or both as being detrimental[11–13] while some studies did not show a significant relationship with sleep duration and cognition[14,15]. The strength of the inferences from these studies have been limited, however, due to small sample size or relatively insensitive cognitive measures. A further issue involves the classification of 'short' and 'long' sleep durations with studies choosing variable thresholds ranging from less than five to eight hours as being 'short' while other studies consider greater than seven to 10 h as 'long' sleep durations. This block duration approach imposes a linear relationship between cognition and more or less sleep around a chosen cut-off.

Sleep duration has also been investigated as a risk factor for dementia. A recent longitudinal analysis of a 7,959 sub-cohort of the Whitehall study suggested that low sleep duration of six hours or less was associated with a higher dementia risk compared to sleeping seven hours, however a long sleep duration of eight hours or more was not[16].

The relationship between sleep duration and brain structure in middle-to-late life has been explored by only a handful of studies. A whole brain approach identified that longer sleep duration was associated with a thinner cortex in the left inferior occipital gyrus[17], while shorter sleep duration related to smaller orbito-frontal cortex and precuneus[18]. One study specifically examined hippocampal volume––often used as an important marker of memory function and Alzheimer pathology––and found no association with sleep duration but did show greater rate of volume loss with poor sleep quality and efficiency[19]. Longitudinal data showed increased rate of cortical thinning in the superior temporal gyrus, inferior and middle frontal gyrus with shorter sleep duration and superior frontal gyrus with longer sleep duration[17], as well as in frontal and temporal regions with poor sleep quality[20].

Notwithstanding these positive findings, the results of a recent investigation examining 613 participants did not find an association between sleep duration and overall grey and white matter volume and further longitudinal assessment, using latent class growth analysis, suggested the trajectory of sleep across 25 years similarly did not impact brain structure[21]. Several reasons may explain the heterogeneity of these findings. A major challenge is low sample size which reduces power to detect significant change, especially in longitudinal studies. Further, a pre-specified region of interest approaches may miss important areas of the brain related to sleep.

To summarise, there remain several important questions around sleep and brain health. To the best of our knowledge, the literature lacks a clear parametric visualisation of the relationship between sleep duration and cognition across age in later years. As a result we cannot answer the following questions with any degree of conviction: Is there an optimal sleep duration for healthy middle-to-late life individuals and does this change as we grow older? What is the relationship between sleep duration and brain structural health, and if one exists, how widespread is this effect?

Here, we have investigated data from the UK Biobank, a large population cohort of middle-to-late age individuals who underwent medical, lifestyle and genetic assessment. We examined the relationship between sleep duration and cognitive performance in 479,420 healthy individuals between 38 and 73 years of age while considering the effect of socioeconomic factors, cardiovascular and genetic risk. In the most extensive neuroimaging study with sleep data, we also explored the relationship between sleep and brain volume in a sub-group of 37,898 individuals who attended for multi-modal brain imaging. Given our large sample size, a whole brain approach was possible to assess the nature of the potential effect of sleep on brain structure.

## Results

**Population characteristics of 479,420 healthy middle-to-late age individuals**. Demographic, clinical and genetic data was analysed in 479,420 individuals from the UK Biobank aged between 38–73 (mean age 57, SD 8) years at the time of their assessment (Fig. 1 and Supplementary Table 1). Self-reported daily sleep duration was a main sleep variable of interest (median seven hours). Additional sleep characteristics included chronotype (27.0% identified as a "morning person", 8.9% "evening person" and 64.0% were intermediate), report of insomnia (24.2% responded "never/rarely", 47.8% "sometimes" and 28.0% "usually") and obstructive sleep apnoea traits including report of snoring (58% "no", 34.5% "yes") and daytime sleepiness (75.8% "no", 20.8% "sometimes" and 2.7% "often").

One percent of the cohort was diabetic, 13% were hypertensive and 18% had high cholesterol, based on medication history. The average body-mass index (BMI) was 27.4 and 77% of the cohort were classified as non-smokers. The apolipoprotein-E (*APOE*) allele frequency within the cohort was 69% with ε3/3 allele, 27% with one ε4 allele and 1% having ε4/4 genotype, based on individuals with available data. Further analysis was performed on a subset of 37,553 individuals who underwent an additional assessment with magnetic resonance imaging (MRI) of the brain aged between 45 and 83 years (mean 65, SD 8).

**Relationship between age, sleep duration and executive function**. A continuous cognitive function latent variable of Executive Function was estimated using confirmatory factor analysis (CFA) from five cognitive tasks of working memory or speed of processing (model and fit indices shown in Fig. 1c, d). Sliding window analysis showed a clear decline in Executive Function with increasing age (Fig. 2a, b), consistent with previous findings using a similar measure[22]. Sleep duration remained stable across age around a median of seven hours (Fig. 2c). Age has been shown to be the most significant confounder affecting cognition. We therefore controlled for the age effect using a quantile-based, age-residual analysis: each person's cognition was expressed relative to the mean of people with matched age quantiles. Then these age residuals were plotted for Executive Function against discrete sleep durations which revealed that seven hours of sleep was associated with the highest Executive Function score (Fig. 2d).

Each shorter sleep duration from six to three hours corresponded with increasingly worse performance while a similar negative effect was seen from eight to 12 hours of sleep, reflecting a quadratic relationship. In a sensitivity analysis examining the association between sleep duration and performance on each individual cognitive task that was used to create the Executive Function latent variable, the same quadratic relationship was observed in all but one task (Supplementary Fig 1).

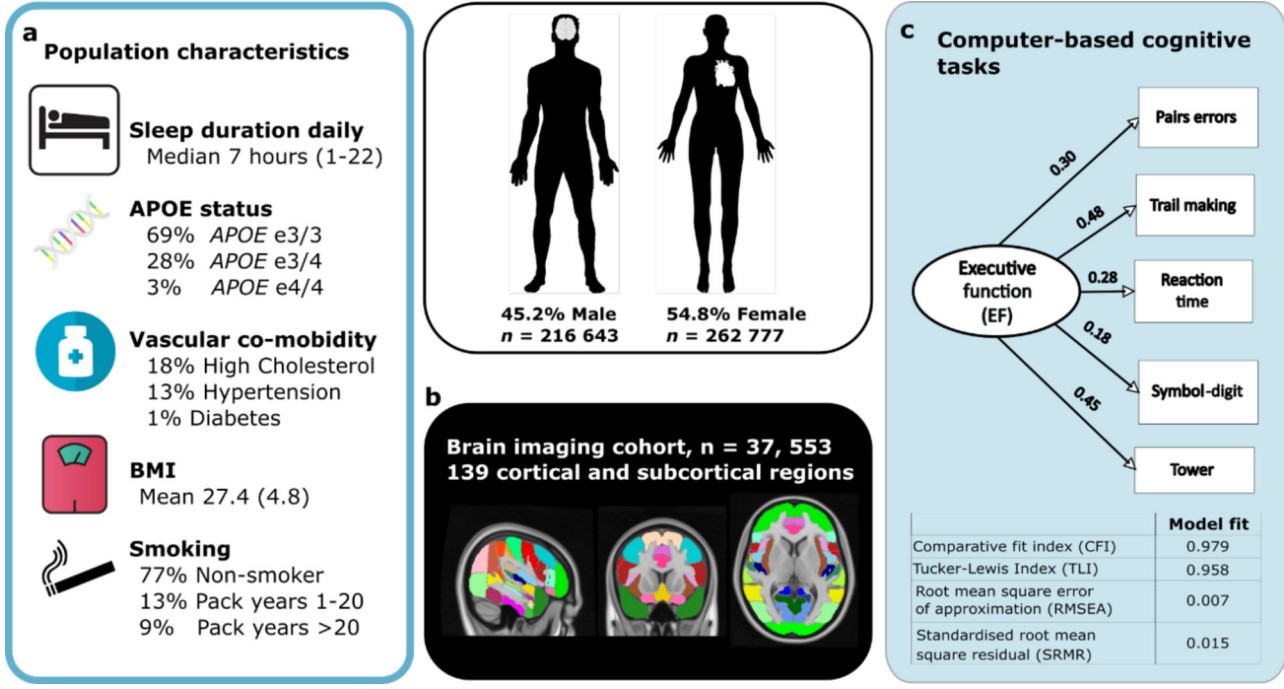

**Fig. 1 Population characteristics for 479,420 healthy middle to older individuals in UK Biobank, executive function score and brain imaging. a** Population characteristics including self-reported sleep duration, vascular co-morbidities, body mass index (BMI), smoking history and *APOE* genotype. **b** Magnetic resonance imaging (MRI) in subset of 37, 553 individuals using a whole brain approach to estimate grey matter volume in 139 cortical and subcortical regions. **c** Executive Function score calculated from performance on five working memory or speed of processing computer-based tasks using confirmatory factor analysis, values show beta-estimates. Model fit indices for factor analysis shown below. *APOE* apoprotein E; BMI body mass index.

The pairs-matching task did not show a clear quadratic relationship with sleep which may reflect a performance ceiling effect. This sensitivity analysis therefore indicates that the individual tasks were well represented by the latent variable.

**Sleep and executive function in middle-and-late age participants.** We next examined the relationship between sleep duration and Executive Function age-residuals in younger (38–59 years, $n = 262,409$) and older (60–73 years, $n = 212,006$) participants, with this age boundary being close to the median age of the group (58.6 years) and corresponding to previous literature[23,24]. Crucially, the pattern of the relationship between sleep duration and Executive Function observed in the entire cohort remained present in both groups and, when controlling for age, seven hours of sleep was still associated with the highest Executive Function score (Fig. 2e, f). The effect of sleep on cognition was similar in both age groups, but in the older group was associated with a smaller variance and range of Executive Function scores relating to sleep duration as compared to the younger group, $F$ (262,409, 212,006) = 1.01, 95% CI [1.0017, 1.0181], $p = 0.017$). Furthermore, the negative effect of very short (two three hours) and long (11 & 12 hours) sleep durations were more apparent in the younger group.

**Sleep duration predicts Executive Function in addition to cerebrovascular and genetic risk.** Multiple regression with Executive Function being the dependent variable was performed to examine the effect of six-to-eight hour sleep duration compared to other reported sleep times, reflecting a quadratic relationship (Table 1). The regression model controlled for age, sleep characteristics including chronotype and report of insomnia, obstructive sleep apnoea traits (daytime sleepiness and snoring) as well as vascular co-morbidity, smoking, BMI, APOE ε4 genotype and socioeconomic status. This sleep band was chosen based on the seven hour sleep duration associated with the highest cognitive performance in addition to one hour on either side to account for self-reporting bias. Age was the strongest predictor by an order of magnitude compared to other predictors, and further emphasised the need to regress out age confounding effects in earlier analyses. Smoking status and number of vascular comorbidities (considering hypertension, hypercholesterolaemia and diabetes) were significant predictors as well as the *APOE* ε4 genotype and socioeconomic status. Daytime sleepiness, an obstructive sleep apnoea trait, was a significant negative predictor of Executive Function. Importantly, sleeping between 6-8 hours remained a significant, and only positive, predictor of Executive Function after controlling for these other factors.

**Larger brain volumes in individuals who sleep between six-to-eight hours.** Whole brain analysis was performed using T1 volumetric MRI data in 37,553 individuals of the UK Biobank cohort. The brain was divided into 139 cortical and subcortical regions of interest based on the Harvard-Oxford Atlases. To investigate a potential 'optimal' sleep duration, we first compared the mean grey matter volumes of all individuals who slept between six-to-eight hours with that of individuals who reported other sleep durations.

Forty-six brain regions were significantly larger in volume in the six-to-eight hour sleep group (following permutation testing with Bonferroni correction for multiple comparison), including the orbital frontal cortex, pre-and post-central gyri, right frontal pole, hippocampi and thalami (Fig. 3a). Several significant brain regions were cerebellar sub-regions (21 out of the 46) which reflected the extensive cerebellar coverage of the atlas. The significant differences in volume ranged from 2.0% larger (in the right nucleus accumbens) to 0.5% larger in the right frontal pole. There were no smaller brain regions in the six-to-eight hour sleep duration group.

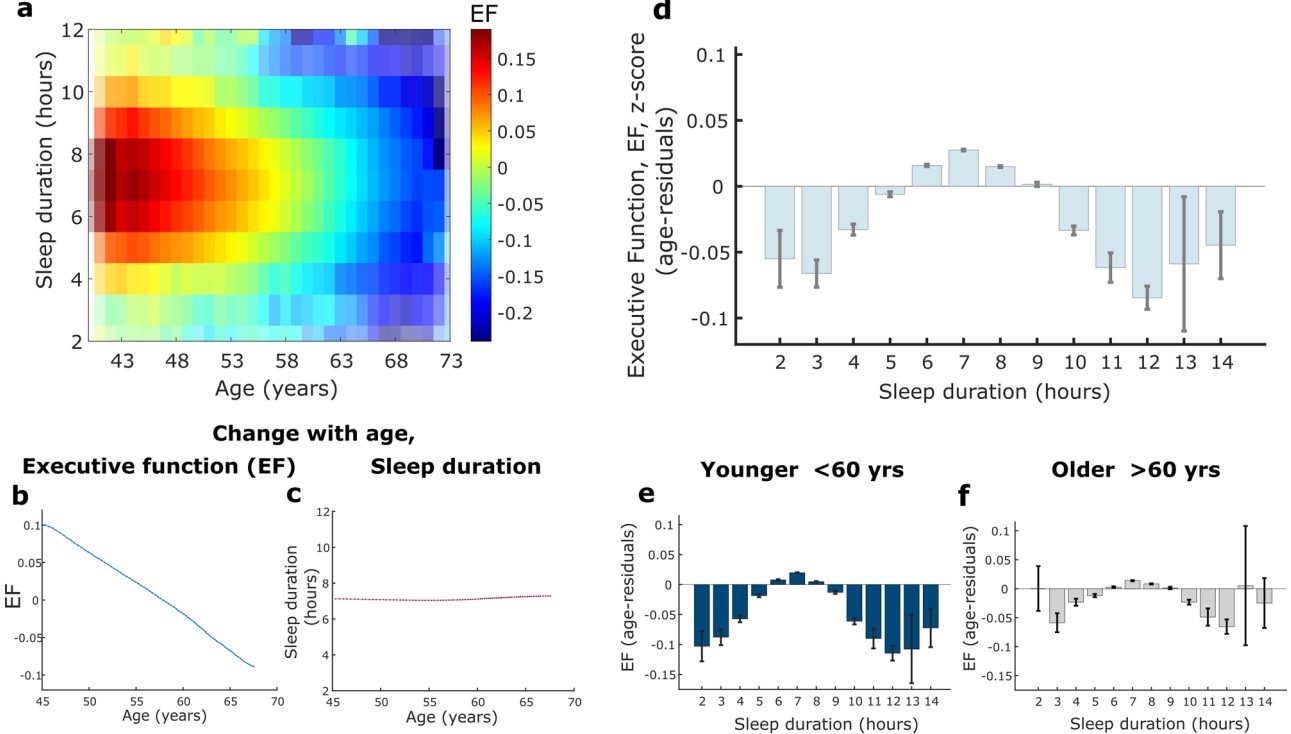

**Fig. 2 Relationship between age, sleep duration and Executive Function. a** Younger age was associated with a higher Executive Function (EF) score, reflected by colour intensity, which peaked between six to eight hours sleep duration (EF standardised into z-score, n = 474, 417, transparency for colourmap relates to number of datapoints). **b**, **c** EF score decreased with as a linear function of age while average reported sleep duration remained constant with age around a median of 7 hours (shaded error represents standard error). **d** Association between sleep duration and EF score, adjusting for age (using an age-residual approach), showed that seven hours of sleep was associated with the highest EF score. A negative relationship was present with less sleep from six to three hours as well as more sleep from eight to 12 hours (**d**, **e**, **f**: error bars represent standard error). **e**, **f** The relationship between age-residual EF scores and sleep duration in younger (38–59 years, n = 262, 409) and older (60-73 years, n = 212, 006) participants followed the same trend as the whole group with the highest EF score still associated with seven hour sleep duration. However, the group variance and range was significantly smaller for older participants, $F(262409, 212006) = 1.01$ (95% CI [1.0017, 1.0181], $p = 0.017$). EF Executive Function, CI confidence interval. Boxplots for Fig. 2d, e, f are shown in Supplementary Fig 6.

**Table 1 Sleep, genetic, socioeconomic and cardiovascular predictors of executive function.**

| Variable | Unstandardized beta estimate | Standard error | t | p |
|---|---|---|---|---|
| Intercept | 0.752 | 0.026 | 29.30 | *<0.001 |
| Age at assessment | −0.013 | 0.001 | −43.23 | *<0.001 |
| Sleep band 6–8 h | 0.027 | 0.008 | 3.45 | *<0.001 |
| Sleep chronotype | | | | |
| Intermediate (baseline) | – | – | – | – |
| Morning | −0.021 | 0.005 | −4.40 | *<0.001 |
| Evening | 0.014 | 0.007 | 1.89 | 0.059 |
| Report of insomnia | 0.002 | 0.003 | 0.84 | 0.400 |
| Obstructive sleep apnoea traits | | | | |
| Daytime sleepiness | −0.22 | 0.004 | −5.48 | *<0.001 |
| Report of snoring | −0.010 | 0.004 | −2.35 | *0.019 |
| Smoking pack years | −0.001 | 0.001 | −2.45 | *0.014 |
| Vascular comorbidity | −0.014 | 0.003 | −5.04 | *<0.001 |
| *APOE* ε status | −0.009 | 0.004 | −2.28 | *0.023 |
| Body mass index (BMI) | −0.001 | 0.001 | 1.37 | 0.172 |
| Socio-economic status | −0.010 | 0.001 | −13.77 | *<0.001 |

Using the summary volume measure of these 46 brain regions, sleeping between six-to-eight hours remained predictive of higher brain volume in a regression model which included baseline characteristics of sleep, obstructive sleep apnoea traits, smoking, vascular co-morbidity, APOE ε status, body mass index and socio-economic status (Supplementary Table 3).

When comparing the six-to-eight hour sleep duration group with those who reported sleeping less than six hours or greater than eight hours separately, the above volume difference effect was driven to a larger extent against longer sleep durations (47 significantly larger regions in the six-to-eight hour sleep group, including 24 cerebellar sub-regions). By contrast,

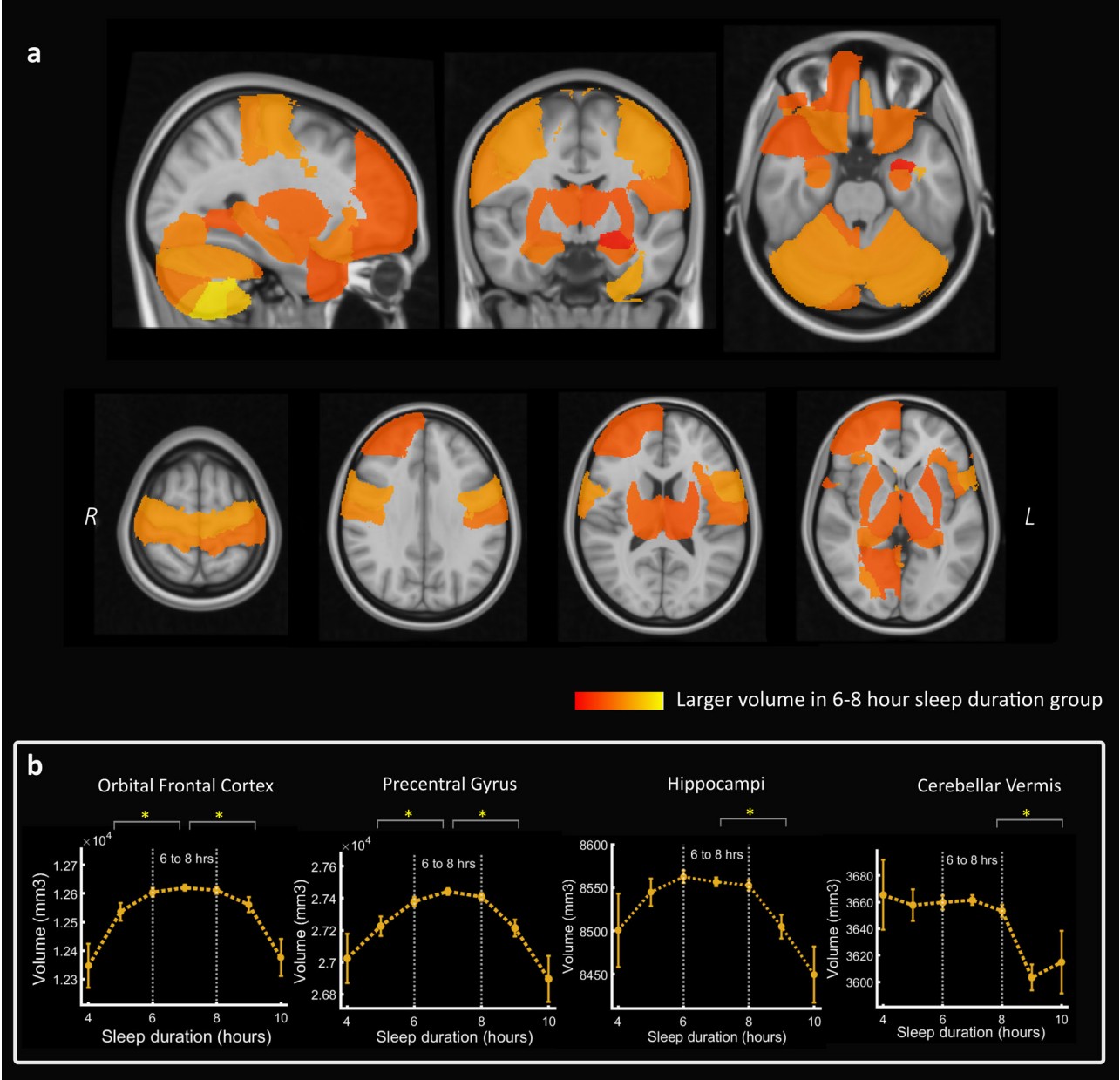

**Fig. 3 Brain regions with a larger volume in individuals who reported six-to-eight hours sleep duration compared to other sleep lengths. a** Volumetric maps showing that 46 out of 139 cortical and subcortical brain regions (based on the Harvard-Oxford atlas) were associated with a significantly higher brain volume in imaging participants who slept between six and eight hours per night compared to other durations ($P < 0.05$ after Bonferroni correction for multiple comparisons). Several frontal, temporal, parietal and cerebellar regions were identified including the orbital frontal cortex, pre- and post-central gyrus, hippocampi, thalami, right frontal pole and anterior parahippocampal gyrus. No brain regions were smaller in the six-to-eight hour sleep duration group. Colormap reflects proportional volume change. **b** Relationship between sleep duration and volume in four selected brain regions showing lower volume associated with both sleep duration extremes for some regions whereas others showed a smaller volume mainly with longer sleep durations (significant difference between sleep group six-to-eight hours compared to less than six or greater than eight hours, *$p < 0.05$ after Bonferroni correction for multiple comparisons, sleep durations with more than 100 imaging participants are plotted). Boxplots for Fig. 3b shown in Supplementary Fig 6.

the six-to-eight hour sleep group had a larger volume in seven regions compared to individuals with the shorter sleep durations; in the orbitofrontal cortex and precentral gyrus bilaterally, right frontal pole, right posterior cingulate and right amygdala (see full breakdown of individual region analyses in Supplementary Figs. 2–4). Therefore, several individual brain regions showed a quadratic or inverted 'u'-shape relationship between sleep duration and grey matter volume however more areas showed smaller volume with longer sleep duration only (examples in Fig. 3b).

**Relationship between sleep duration and white matter hyperintensity volume.** White matter hyperintensity (WMH) volume, considered a structural marker of cerebrovascular burden on the ageing brain[25], was plotted against sleep duration. Eight hours of sleep was associated with the lowest burden of WMH (Fig. 4a) and individuals who slept six-to-eight hours had significantly lower WMH volume compared to other sleep durations (following permutation testing with Bonferroni correction). Thus, WMH and sleep duration appear to have a similar quadratic relationship to that seen with other brain structural measures.

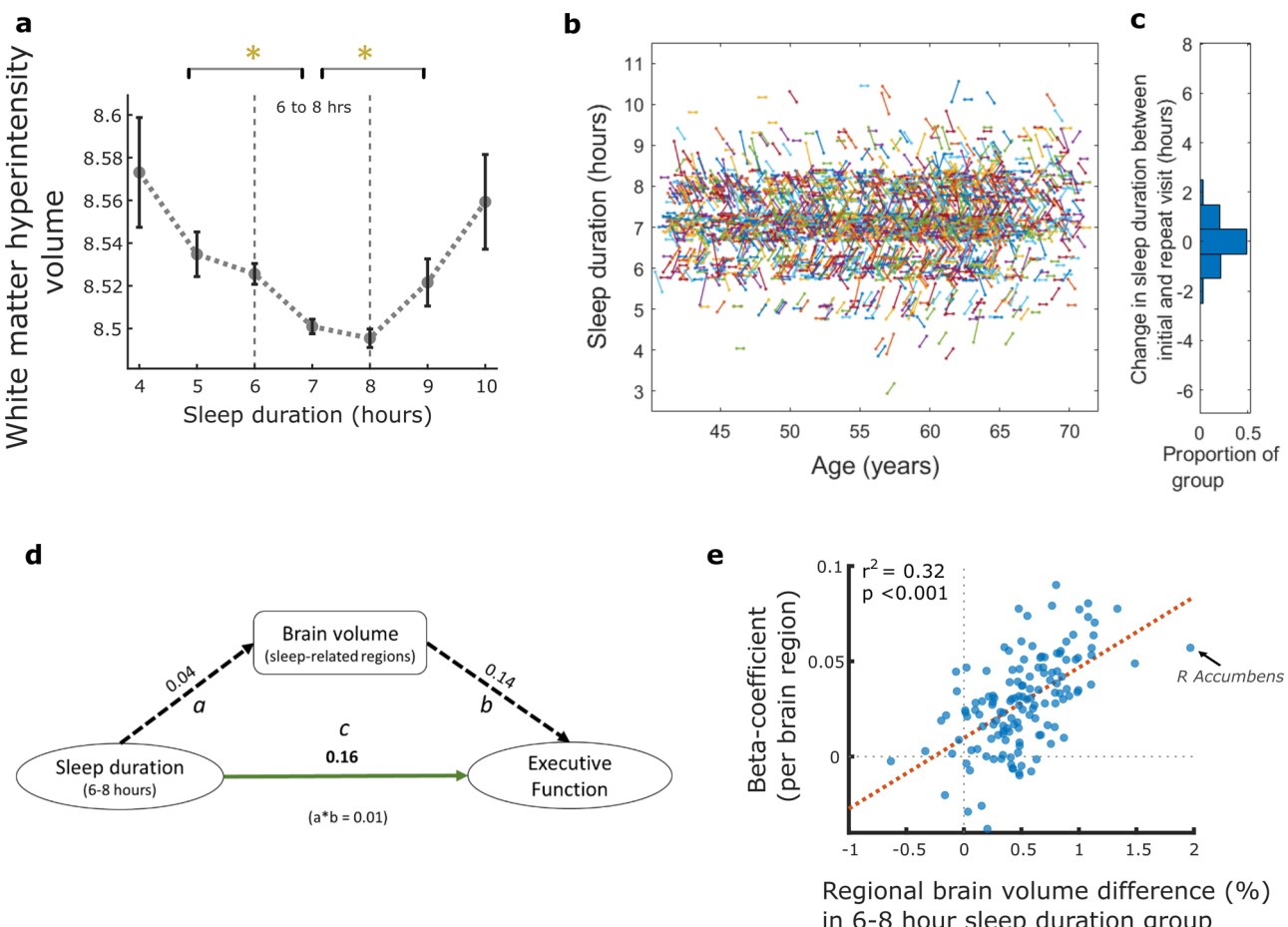

**Fig. 4 White matter hyperintensity load and sleep duration, sleep trajectory and relationship between sleep duration, brain volume and Executive Function. a** Association between sleep duration and white matter hyperintensity volume (WMH), log-transformed, showing the lowest WMH burden with eight hours of sleep. Those who slept for durations other than six-to-eight hours had significantly greater burden, $P < 0.05$ following Bonferroni correction for multiple comparison (error bars denote standard error). **b** Sleep trajectory plotted for a random subset of the imaging cohort (5%, $n = 2,416$, for visualisation purposes) who provided sleep data at two timepoints, mean 8.9 (sd 1.8) years apart. Each line represents an individual, plotted by age, with the gradient reflecting any change in sleep duration between initial and imaging assessments (**b**: sleep axis points have been jittered by 5% to help distinguish clusters). Trajectories were mainly flat or had a shallow gradient as reflected in **c**, the distribution of sleep duration change for the entire imaging cohort ($n = 37, 170$), as almost half the group (48.5%) reported no change in sleep duration while 91.0% reported either the same sleep duration or a difference of only one hour. **d** Mediation analysis, values are standardised beta estimates. Path a: relationship between sleep duration of six-to-eight hours and sleep-related brain region volume. Path b: relationship between sleep-related brain region volume and Executive Function. Path c: direct relationship between sleep duration (6–8 hours) and Executive Function (a*b value is the beta estimate for indirect pathway whereby brain volume mediates the relationship between sleep and Executive Function). All paths were significant at $p < 0.001$. **e** Correlation analysis showing that higher regional brain volume in participants with six-to-eight hour sleep duration correlated with the strength (beta-coefficients) of regional brain volume predicting Executive Function ($r2 = 0.32$, $*p < 0.001$). Each datapoint is a brain region (example arrow indicating right Accumbens). Boxplots for Fig. 4a shown in Supplementary Fig 6.

**Stable sleep trajectory within the imaging cohort.** Individuals who attended for MRI provided a second datapoint with repeat clinical and demographic information, including sleep. The mean time interval between baseline and imaging assessment was 8.90 years (SD 1.76) within the group. We calculated a sleep trajectory based on these two timepoints (Fig. 4b, c) and showed almost half the group (48.5%) reported no change in sleep duration, while 91.0% reported either the same sleep duration or a difference of only one hour. These findings therefore show a relatively stable sleep duration between the two visits for these participants.

**Relationship between sleep and executive function is partially mediated by brain volume.** Mediation analyses was used to further examine the relationship between sleep duration, brain volume and executive function (Fig. 4d). Sleeping between six-to-eight hours was significantly associated with higher executive

function (direct path c) and this relationship was partially mediated by brain volume, using the summary volume measure of sleep-related regions, whereby the relationship between sleep duration and executive function reduced with the inclusion of an indirect path between sleep duration and brain volume (path a) and brain volume to executive function (path b). This reduction was represented by a modest, but significant, drop in beta value of a*b = 0.01 ($p < 0.001$).

**Brain areas associated with six-to-eight hour sleep duration were also the ones that best predict executive function.** A comparison of individual brain regions associated with both sleep duration and executive function was performed. Volume in 56 brain regions significantly predicted executive function (Supplementary Data 2) while volume in 46 brain regions showed a significant quadratic relationship with sleep duration,

as shown above. Forty-one of these brain regions overlapped (Supplementary Fig. 5) including hippocampi, thalami, orbito-frontal cortex, right frontal pole, left parahippocampal gyri and cerebellar regions. Further, regional brain volume differences (%) between individuals who slept between six-to-eight hours compared to other durations were correlated (r2 = 0.32, *p < 0.001) with the strength (beta-coefficients) of regional brain volume predicting executive function (Fig. 4e). This showed that regions which were larger in individuals who slept six-to-eight hours were also larger in those with high executive function. Conversely, regions which did not relate to sleep in this way, also did not predict executive function.

## Discussion

The results of this study show that the amount of time people sleep, an important and potentially modifiable lifestyle factor, is predictive of cognitive function across age in middle-to-late healthy individuals. Peak cognitive performance was associated with seven hours of overnight sleep in the sizeable cohort of the UK Biobank. In the largest neuroimaging study to date, sleeping between six-to-eight hours was associated with greater grey matter volume in a distributed anatomical pattern involving frontal, temporal, parietal and cerebellar regions.

Our findings highlight the important relationship between sleep duration and cognition and provide a meaningful visualisation of the parametric effect for every hour of sleep on a latent variable of Executive Function across age. These findings expand on a previous study which associated sleeping too long (10 h or more) and too short (six hours or less) with worse performance in some UK Biobank cognitive tests[26], and helps to resolve earlier mixed findings in the literature which used variable cut-offs for deciding what constitutes short, long and baseline sleep durations. Of practical importance, our findings suggest that every hour around an optimal sleep duration may relate to differences in cognitive performance and, in contrast to previous discussions[11,21], avoiding very short and long amounts of sleep is not enough.

Further age-group analysis showed that the association between sleep duration and cognition was greater in a younger group of this cohort but, importantly, remained present in older participants (>60 years). While the cross-sectional nature of this analysis was limited against causal inference, our findings in combination with a recent a mendelian randomisation study looking at sleep and dementia risk in the same cohort[27], suggests an optimal sleep duration does exist to impact our daily cognitive function. This has implications for every adult as an important modifiable lifestyle factor for healthy cognitive ageing, especially considering the natural decline across our cognitive domains as we grow older[28] and the global ageing population[29].

Sleep duration has a significant relationship with brain structural health. Our findings identified frontal, temporal, parietal and cerebellar regions with a larger grey matter volume in individuals who had six-to-eight hours of sleep compared to other sleep durations. Identifying these regions was made possible by the large sample size in the UK Biobank and offers two important observations which help to understand disparate findings from the literature. Previous cross-sectional studies in middle-to-older individuals have linked extremes of sleep duration with either ventricular enlargement[6], smaller orbitofrontal cortex and precuneus volume[18], thinner inferior occipital gyrus[17] or no specific association with frontal or hippocampal volume[6,30]. Our results indicate a far more widespread relationship exists between brain structural integrity and sleep duration and is more consistent with longitudinal data associating increased rate of atrophy in frontal, parietal and temporal regions with poor sleep quality[20].

In our cohort, individuals with longer sleep durations had a smaller cerebellum which has not been described in healthy middle-to-older individuals, although low cerebellar volume has been associated with poor sleep quality in adolescents[31], and with abnormal functional imaging with sleep disruption conditions such as narcolepsy and obstructive sleep apnoea[32]. Secondly, we demonstrated a quadratic relationship between sleep duration and brain volume, similar to sleep and cognition, in several brain regions while other areas showed lower volume with longer sleep durations only. While a close association between sleep, cognition and brain structure may appear intuitive, this has not been previously been shown in a single cohort, and was supported by mediation analysis findings and the large overlap of individual brain regions associated with sleep and cognition. While causal pathways cannot strictly be inferred, our findings highlight the important relationship between sleep duration, cognition and a large structural network of brain regions.

In addition to structural grey matter integrity, a smaller volume of white matter hyperintensities was observed in individuals who slept between six-to-eight hours. Considered as a brain marker closely related to cardiovascular burden[33], this further implicates sleep duration as an important factor within the interplay of cardiovascular risk impacting brain health.

The quadratic, or 'u'-shaped', relationship between sleep duration with cognitive function and some brain measures has both practical and biological considerations. While some national guidelines suggest that adults should get seven hours of sleep or more, with no upper boundary[34], our findings are more in line with guidance for a target range of seven to nine hours of sleep[35]. From a biological perspective, it is likely that more than one biological mechanism underpins our findings. Shorter sleep durations have been linked with reduced density of slow-wave sleep activity in fronto-temporal regions with associated cortical thinning[36], possibly related to excessive wakeful neuronal activity[37,38].

Recent findings suggest that even a single night of sleep deprivation can limit clearance of neurodegenerative proteins from the brain[39] and chronic short sleep durations may lead to accumulation of such proteins with resultant effect on brain function[40,41]. By contrast, very long reported sleep durations may reflect other problems of mood-related chronic illnesses or hypersomnolence disorders such as obstructive sleep apnoea, which can impair cognitive processes directly or indirectly through poor sleep quality or sleep fragmentation[42]. While a mechanistic understanding is not easily obtained from this dataset, our findings emphasize the need for careful evaluation of sleep habits and difficulties if an individual consistently falls outside a healthy sleep duration range for both cognitive function and brain structural health.

There are several important limitations to this study. We mainly analysed cross-sectional data from the UK Biobank cohort who have a higher socioeconomic baseline and fewer comorbid disease compared to the general population[43]. The self-reported nature of sleep duration may lead to reporting inaccuracy and recall bias with a previous study suggesting that older adults report sleep durations longer than objective measurements, especially when sleep quality was poor[44]. Our subjective sleep durations were reported in discrete one hour blocks resulting a relatively insensitive measure which included daytime naps. To mitigate these issues, we visualised the relationship between every hour of sleep and variables of interest, wherever possible, or considered a sleep window around the optimal sleep duration observed from initial analysis. Despite including report of insomnia in our regression model, our study could not reliably assess sleep quality or the specific nature of sleep, such as the proportion spent within different sleep stages, which will impact

the effect on cognitive function[45]. However, the UK Biobank is currently releasing actigraphy data which will improve the resolution of sleep duration for future studies and add valuable information about sleep quality.

The findings presented here provide insight on the relationship between sleep and the integrity of a widespread structural brain network. We also demonstrate how the modifiable lifestyle factor of sleep duration is related with cognitive performance which have implications on the everyday routine of middle-to-late life individuals in maintaining healthy cognitive ageing.

## Methods

**Participants.** Data from 479,420 people aged between 38 and 74 years of age from the UK Biobank were analysed including detailed health, demographic and cognitive assessments. MRI data from 37,533 people was used for our imaging analysis. Information from the initial baseline visit, performed between 2006 and 2010, and the imaging visit, acquired between 2014 and 2020, as well as follow-up online questionnaires and hospital record data was used. All participants provided written, informed consent and the study was approved by the Research Ethics Committee (REC number 11/NW/0382).

We excluded individuals with history or current diagnoses of neurological disease, stroke, transient ischaemic attack, brain injury, subdural or subarachnoid haematoma; infection of the nervous system; brain abscess, haemorrhage or skull fracture; encephalitis, meningitis, chronic neurological problem, amyotrophic lateral sclerosis, multiple sclerosis, Parkinson's or Alzheimer's disease, epilepsy, head injury, alcohol, opioid and other dependency according to the non-cancer illnesses codes (http://biobank.ndph.ox.ac.uk/showcase/coding.cgi?id=6) and algorithmic-defined outcomes (https://biobank.ndph.ox.ac.uk/showcase/label.cgi?id=42).

**Cognitive testing.** Cognitive testing in the UK Biobank was performed at the initial baseline visit and imaging visit, administered via touchscreen, and through an additional online questionnaire. Not all cognitive tasks were done at each instance while some were repeated. We analysed data from five tasks of working memory or speed of processing, to index executive function, and used the first available timepoint data. We used the number of errors made in the pairs matching task, where participants had to memorise the position of six matching card pairs presented simultaneously and identify the location of these pairs after the cards were turned over. The Trail-making task involved linking consecutive numbers (numeric version A) or alphabets and numbers (alphanumeric version B) sequentially. We calculated the difference between completion time for both task versions, an index of executive function[46]. Accuracy at the Tower Rearranging task, a variation of the Tower of London working memory task, was calculated. In each round, participants were shown three pegs ('towers') which had three different coloured rings placed and were asked to indicate the number of moves required to re-arrange the hoops to specific location. Reaction time was indexed from a variation of the card game, Snap. We calculated accuracy in the Symbol-Digit Substitution task where participants had to match symbol-digit codes to test set of symbols. Reliability and re-test effect over time for these cognitive tasks have been previously assessed[47].

**Sleep duration, cardiovascular, genetic and socioeconomic factors.** Sleep duration was a key variable for this study. Individuals reported the number of hours they sleep every 24 h. This information was obtained during the UK Biobank UK assessment and again for individuals who attended the imaging visit. Other sleep characteristics considered included chronotype with choices of "definitely morning", "definitely evening", "more morning than evening" and "more evening than morning". Similar to previous work[26], we created a categorical chronotype variable by combining participants who chose the latter two options as the baseline compared to "morning" or "evening" individuals. Report of sleepiness/ insomnia and was scored ordinally based on responses of: "never/ rarely", "sometimes" and "usually. Obstructive sleep apnoea traits were ordinally based by responses of "yes" and "no" for snoring (self- or partner-report) and "no", "sometimes" and "often" for daytime sleepiness. We identified several cerebrovascular risk factors shown to affect cognitive function and brain structural health across ageing, including in this cohort[22] (Supplementary Table 1). Self-reported hypertension, hypercholesterolaemia and diabetes contributed to a cardiovascular co-morbidity rating (maximum of three points). Body mass index was recorded during baseline assessment and smoking status reflected pack year history calculated from current and previous smokers and amount reported (one pack year equated to smoking 20 cigarettes daily for one year). For previous smokers, this was adjusted to the age an individual stopped smoking.

In terms of a genetic risk, we considered the APOE ε4 allele status based on known effect on cognition and risk of developing sporadic dementia. To score this risk, a point was given for every ε4 allele an individual carried (one point for heterozygous and two for homozygous carriers), while carrying APOE ε3/ε3 did not confer any points and other ε alleles were not considered in this study.

Genotyping was conducted by Affymetrix for UK Biobank using bespoke Axiom arrays[48]. Socioeconomic status was estimated by the Townsend Deprivation Index which is a measure of deprivation based on unemployment, home and car ownership and household overcrowding according to participant postcode.

**MRI data acquisition, pre-processing and analysis.** Imaging data were acquired on a Siemens Skyra 3 T scanner with a 32-channel head coil. The full imaging protocol including acquisition details is openly available (https://biobank.ctsu.ox.ac.uk/crystal/docs/brain_mri.pdf). High-resolution (1 mm isotropic voxel), T1-weighted, 3D magnetization-prepared gradient echo structural images and a T2 weighted fluid-attenuated inversion recovery (FLAIR) images (1.05 mm × 1 mm × 1 mm resolution) were obtained as part of a longer MRI protocol. Preprocessing and quality checking of images followed a standardized and openly available pipeline with details published elsewhere[49,50]. Important to this study, key stages of the T1 processing pipeline included gradient distortion correction and registration to MNI152 space, segmentation into grey matter, white matter and cerebrospinal fluid using FAST (FMRIB's Automated Segmentation Tool), and then bias field correction[50]. Subcortical structures were further modelled using FIRST (fMRIB's Integrated Registration and Segmentation Tool) which generated 15 subcortical structures. The FLAIR image was also gradient distortion, bias field corrected and linearly registered to the T1 image and to Montreal Neurological Institute (MNI 152) Atlas space. WMH segmentation was carried out using an automated method for classifying voxels based on relative intensity and spatial features called the Brain Intensity Abnormality Classification Algorithm (BIANCA)[51].

Our analysis was based on summary statistics of key brain imaging variables, known as imaging derived phenotypes (IDPs), of the cortical and subcortical grey matter volume estimates and WMH volume. The grey matter IDPs were parcellated according to the Harvard-Oxford cortical and sub-cortical atlases which resulted in 139 brain regions, including lateralised and midline structures (https://biobank.ndph.ox.ac.uk/showcase/label.cgi?id=110). We used 12 subcortical regions that was generated using FIRST which overlapped with the FAST segmentation output because we reasoned that FIRST has been specifically optimised for deeper brain structures[52] that often have poorer imaging contrast.

We used the median absolute deviation to identify and exclude outliers based on previous work on the imaging dataset[53]. Measures of age, age*2, head size, table position and scan-date-derived confounds were regressed out using a General Linear Model approach. The squares of ages accounted for quadratic dependencies of IDPs on age. Head size was a scaling factor based on the transformation of the individuals structural MRI to the standard template space. This method of addressing confounds has been extensively investigated[50] and these variables were chosen as most relevant to our question or important confounds for this dataset, as previously considered[54].

**Statistics and reproducibility.** Calculations were performed in Matlab R2018a or in R, using the Lavaan package[55], for confirmatory factor analysis (CFA) and mediation analysis. Cognitive variables were pre-processed to correct for heavily skewed distribution prior to CFA. Standard fit indices were measured with higher comparative fit index (CFI) and Tucker-Lewis Index (TLI) considered better (>0.9 is commonly used as an acceptable fit cut-off) while lower root mean square error of approximation (RMSEA) and standardised root mean square error residual (SRMR) is considered better (<0.06 and <0.08, respectively, are commonly used cut-offs for acceptable fit). Reaction time, trail-making difference was log transformed while pairs-matching errors was cube root transformed. We estimated a latent variable from the five cognitive tasks described above using CFA, which has the methodological advantage of controlling measurement error that can artificially reduce the relationship between measured variables in standard univariate analyses[56]. Performance in these tasks generally share covariance and show steep declines with age[28], hence we reasoned that they may index a common latent variable. This approach has the benefit of creating a continuous summary representation from multiple measures of working memory and speed of processing which we termed, for simplicity, "Executive Function". We reported standard CFA model fit indices. Missing data was estimated using full information maximum likelihood, which gives unbiased parameter estimates and standard errors. To address any concerns about bias in the missing data, we also performed sensitivity analyses on performance from individual cognitive tests and the primary variable of interest, sleep duration, for comparison with our Executive Function latent variable.

The relationship between sleep duration and the Executive Function latent variable across age was first visualised using a heatmap ($N = 479,420$). A mean smoothing factor with one hour on either side of each sleep duration and two years around each age year was applied. A sliding window approach was then used to study the individual relationship between age and Executive Function and between age and sleep duration. This method does not assume a linear relationship between variables. An age window of observations of fixed age-quantile widths was moved along the age distribution (code: conditionalPlot.m available here: https://osf.io/vmabg/)[53,57] with a smoothing Gaussian kernel of five applied across each window. This method has previously been used on this dataset[22,53] and code is openly available[57]. Deconfounding age from our Executive Function measure was done using age residuals within fixed-age quantile bins of 20% (7.1 years) and plotted

against sleep duration for the entire group and for subgroups of younger (<60 yrs, $N = 264,935$) and older (>60 yrs, $N = 212,006$) participants. Differences in the distribution between these age sub-groups was assessed with a two sample F-test for equal variance.

Multiple regression was used to test the relationship between sleep duration and the Executive Function latent variable while controlling for age, cardiovascular comorbidity, smoking pack years, BMI, genetic risk and socioeconomic status. Participants within a six-to-eight hour sleep duration scored one point, while others scored zero, to reflect the quadratic nature between sleep duration and Executive function identified during initial analysis and previously reported[26]. The regression model was tested for multicollinearity using variance inflation factor. We set the threshold for statistical significance at $p < 0.05$.

To investigate the association between sleep duration and brain volume, we first compared individuals who sleep six-to-eight hours and those with other sleep durations ($N = 37,553$). WMH volume was log-transformed prior to analysis due to the skewed nature of the data distribution. The difference in each mean brain area volume was tested using a permuted t-statistic, applying a Bonferroni correction for multiple comparisons, corrected $p$ value = 0.05/140 (to account for 139 brain regions and white matter hyperintensity volume). We chose this 'strict' correction approach due to the number of brain areas involved. Permutation testing was done by shuffling datapoints for each brain region for the two different sleep duration groups. We then tested 1000 permuted datasets per brain region and calculated the p-value as the proportion of permutations that resulted in a t-statistic that was larger or equal to the observed one in the original data. The difference in brain volume was presented as a percentage based on the average of the two groups and significant brain regions were visualised using a fMRIB 'atlas-fill' python module (https://git.fmrib.ox.ac.uk/thanayik/atlas-fill), superimposed onto a standard MNI template image. A summary volume measure was calculated for brain regions with a significant quadratic relationship with sleep, described as "sleep-related brain regions", by performing a z-score transformation of regional volume and taking the average across regions. This was used as the dependent variable in a regression model to account for other covariates described above (sleep characteristics, obstructive sleep apnoea traits, cardiovascular comorbidity, smoking pack years, BMI, genetic risk and socioeconomic status). Permutation testing analysis was repeated separately comparing the six-to-eight-hour sleep duration group with the less than six hour sleep duration group and comparing the six-to-eight hour sleep group with those who slept more than eight hours.

Mediation analyses were conducted to estimate the directional influence of brain volume to executive function. Mediation analysis was run in Lavaan, with nonparametric bootstrapping with 1,000 iterations to estimate direct and indirect effects between variables. Separate linear analyses was identified significant associations between individual brain region volume and executive function. Results were corrected for multiple comparisons using the Bonferroni method and these regions were compared with those associated with sleep duration.

**Reporting summary**. Further information on research design is available in the Nature Research Reporting Summary linked to this article.

## Data availability
UK Biobank data is available to all researchers for health-related research and public interest. Data can be accessed through the Access Management System (details at https://www.ukbiobank.ac.uk/enable-your-research/register). The variables used are detailed in Supplementary Table 2. Source summary data used in Figures are detailed in Supplementary Data 1.

## Code availability
Code for the sliding window analyses is available from https://osf.io/vmabg/.

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

## Acknowledgements
This work was funded by the Wellcome Trust and MRC (Wellcome Trust Principal Research Fellowship to MH, MRC Clinician Scientist Fellowship to SM and Wellcome Trust PhD clinical fellowship to XYT).

## Author contributions
Design, analysis, manuscript writing, critical revisions (XYT); design, analysis and critical revisions of the manuscript (CC, SM, MH).

## Competing interests
The authors declare no competing interests.
