## [Peer Review File · Communications Biology]

Reviewers' comments:

Reviewer #1 (Remarks to the Author):

Authors use UK Biobank to explore the relationship between sleep (self-reported sleep duration), cognition and (in a subgroup) regional brain volume. They show a quadratic relationship between sleep duration and cognition for people below and above 60. This is what would be expected from the literature, but there has been some inconsistency before. The sample size here allows a fairly definitive confirmation of this relationship. As it stands, the message is clear, well-written and useful to publish, but there are a couple of further analyses to be considered.

1) The authors mention that UK Biobank is releasing actigraphy data. I think they have already released actigraphy data in at least 85k people and probably would be worth including here given the limitations of self-reported sleep duration.

2) The relationship between brain volume and sleep duration is interesting. The authors mention sleep apnoea, prevalence of which has been estimated to be as high as 50% of population in men over 65 and may affect brain volume (in cerebellum and elsewhere). Would it be worth exploring links between obesity as a major risk for OSA, sleep duration and brain volume (and are there any other proxy markers of OSA e.g. daytime sleepiness that could be analysed?) .

3) Along the same lines - the authors state there is likely to be more than one biological process affecting the relationship between sleep and cognition/brain volume. However, there are no control variables/sensitivity analyses in the imaging analysis section (not even age – unless I missed it). These covariables could be of interest (at least in the discussion) as they might give a window onto the pathological processes that link sleep with brain volume.

Reviewer #2 (Remarks to the Author):

This work examines the amount of sleep for optimal brain health and function on a very large sample of 479,420 middle aged-older adults from the UK Biobank. The methods and results are clearly presented and provide reliable evidence for a recommended amount of sleep for optimal daytime function.

Specific comments:

1) L101 - Why would a quadratic relationship between sleep and cognitive performance reflect a performance ceiling effect?

2) L123-125 - I assume results for the earlier sections do not take into account the other sociodemographic and lifestyle covariates. If so, could you make this clear here what the covariates are in this same sentence?

3) L172 - Typo - "forty six"

4) L166-196, 190-197 - Were the same sociodemographic and lifestyle covariates used in the Executive Function analyses used here? For consistency and comparability with the cognition measures, these should be added to the model too.

5) The authors have independently shown that sleep affects brain structure and that sleep affects cognition. Have you considered trying to look at potential causal mechanisms, e.g. that short/long sleep is associated with greater atrophy in certain brain regions which mediate greater impairments in cognition? Or perhaps show overlapping brain regions associated with both sleep-related atrophy + greater cognitive impairments?

We would like to extend our thanks to the editor and reviewers for their time assessing our manuscript and providing useful suggestions to improve this work. We have addressed the remarks raised by Reviewer #1 and #2, point by point below.

Substantial changes and new analyses have been done including new covariates to investigate the relationship between sleep apnoea traits with behavioural and imaging outcomes, an additional regression model for brain volume to account for baseline characteristics of vascular, genetic risk and socioeconomic status. New analyses examine a potential mediation effect of brain volume between the relationship between sleep duration and executive function. Changes to sections of the text are shown in red.

Reviewer #1 (Remarks to the Author):

Authors use UK Biobank to explore the relationship between sleep (self-reported sleep duration), cognition and (in a subgroup) regional brain volume. They show a quadratic relationship between sleep duration and cognition for people below and above 60. This is what would be expected from the literature, but there has been some inconsistency before. The sample size here allows a fairly definitive confirmation of this relationship. As it stands, the message is clear, well-written and useful to publish, but there are a couple of further analyses to be considered.

1) The authors mention that UK Biobank is releasing actigraphy data. I think they have already released actigraphy data in at least 85k people and probably would be worth including here given the limitations of self-reported sleep duration.

We thank reviewer 1 for their encouraging comments and description of a clear and useful message from our work.

We agree with the reviewer that actigraphy data would be very interesting to add support to self-reported sleep duration data. However, while actigraphy data has recently been released in an initial subset of UK Biobank participants, this data remains in a raw format. Our research backgrounds in behavioural and imaging analysis do not provide the expertise to preprocess and analyse these data to identify sleep parameters which is a complex topic. In short, we simply do not have the expertise to use these data which are beyond the scope of the study we designed. We thank the reviewer nonetheless for raising this issue and in the Discussion of the revised manuscript we have pointed to the possibility of future studies using actigraphy.

The revised Discussion section includes the following:

The UK Biobank is currently releasing actigraphy data which will improve the resolution of sleep duration for future studies and add valuable information about sleep quality.

2) The relationship between brain volume and sleep duration is interesting. The authors mention sleep apnoea, prevalence of which has been estimated to be as high as 50% of population in men over 65 and may affect brain volume (in cerebellum and elsewhere). Would it be worth exploring links between obesity as a major risk for OSA, sleep duration and brain volume (and are there any other proxy markers of OSA e.g. daytime sleepiness that could be analysed?).

We thank the reviewer for this comment. We agree that obstructive sleep apnoea (OSA) is potentially linked with sleep disruption, cognitive difficulties and changes in brain volume. A recent study examined the link between obesity and snoring (as a marker of sleep apnoea) in the UK Biobank using mendelian randomisation (Campos *et al.*, 2020).

We identified “report of snoring” and “report of daytime dozing or sleepiness” as OSA traits within the UK Biobank. These have now been added to our analysis of executive function and brain volume. As the reviewer correctly predicts, daytime sleepiness was a significant negative predictor of executive function (revised Table 1).

To ascertain the relationship of OSA, sleep duration and regional brain volume, we created a summary volume measure of all brain regions with a significant quadratic relationship with sleep. This measure of ‘sleep-related brain regions’ was used as the dependent variable in a regression model with the same baseline characteristics including OSA traits as well as sleep chronotype, report of insomnia, smoking, vascular comorbidity, APOE ϵ status, body mass index and socio-economic status (Supplementary Table 3). Importantly, sleeping between six to eight hours remained predictive for higher brain volume in these regions. Report of snoring was a significant negative predictor of brain volume.

The revised Methods section now includes the following:

A summary volume measure was calculated for brain regions with a significant quadratic relationship with sleep, described as “sleep-related brain regions”, by performing a z-score transformation of regional volume and taking the average across regions. This was used as the dependent variable in a regression model to account for other covariates described above (sleep characteristics, obstructive sleep apnoea traits, cardiovascular comorbidity, smoking pack years, BMI, genetic risk and socioeconomic status).

The revised Results now includes the following:

Daytime sleepiness, an obstructive sleep apnoea trait, was a significant negative predictor of executive function.

...

Using a summary volume measure of these 46 brain regions, sleeping between six-to-eight hours remained predictive of lower brain volume in a regression model which included baseline characteristics of sleep, obstructive sleep apnoea traits, smoking, vascular co-morbidity, APOE ϵ status, body mass index and socio-economic status (Supplementary Table 3). Report of snoring was a significant negative predictor of brain volume within these regions.

Thank you for bring this issue to our attention.

3) Along the same lines - the authors state there is likely to be more than one biological process affecting the relationship between sleep and cognition/brain volume. However, there are no control variables/sensitivity analyses in the imaging analysis section (not even age – unless I missed it). These covariables could be of interest (at least in the discussion) as they might give a window onto the pathological processes that link sleep with brain volume.

We thank the reviewer for this comment. All imaging variables were deconfounded for measures of age, age squared, head size, table position and scan-date-derived (see methods). These were identified in previous work as important imaging variables to be controlled for.

We recognise, however, that the covariates considered in the sleep and cognition analysis would also be important to include and therefore performed an additional regression model using the summary volume measure of sleep-related brain regions (see reply to comment 2 above). Sleeping between six-to-eight hours remained predictive for higher brain volume in these regions after controlling for these baseline covariates (OSA traits as well as sleep chronotype, report of insomnia, smoking, vascular comorbidity, APOE ε status, body mass index and socio-economic status). This analysis also indicates vascular co-morbidity, smoking and BMI are significantly associated with lower brain volume.

Thank you for your very helpful suggestions and comments.

Reviewer #2 (Remarks to the Author):

This work examines the amount of sleep for optimal brain health and function on a very large sample of 479,420 middle aged-older adults from the UK Biobank. The methods and results are clearly presented and provide reliable evidence for a recommended amount of sleep for optimal daytime function.

We are grateful to the reviewer for their comments and are pleased that the presentation and message of our findings were clear.

Specific comments:

1) L101 - Why would a quadratic relationship between sleep and cognitive performance reflect a performance ceiling effect?

We apologise that this point could have been explained better. There was a quadratic relationship between sleep duration and the executive function latent variable.

Additionally, we analysed the association between sleep duration and the performance on each cognitive task that was used to calculate the latent variable of executive function. All but one task showed the same quadratic relationship. This suggests a consistent finding across measures of working memory and speed of processing, and that the tasks were generally well represented by the executive function latent variable.

For the one task that did not show a convincing quadratic relationship (pairs-matching), this may be due to a performance ceiling effect specifically in this task.

The revised Results now includes:

Each shorter sleep duration from six to three hours corresponded with increasingly worse performance while a similar negative effect was seen from eight to 12 hours of sleep, reflecting a quadratic relationship. In a sensitivity analysis examining the association between sleep duration and performance on each individual cognitive task that was used to create the executive function latent variable, the same quadratic relationship was observed in all but one task (Supplementary Fig 1). The pairs-matching task did not show a clear quadratic relationship with sleep which may reflect a performance ceiling effect. This sensitivity analysis therefore indicates that the individual tasks were well represented by the latent variable.

2) L123-125 - I assume results for the earlier sections do not take into account the other sociodemographic and lifestyle covariates. If so, could you make this clear here what the covariates are in this same sentence?

Thank you for your comment. We have now made the covariates in this regression model clear and also added additional obstructive sleep apnoea traits (as requested by reviewer #1). The revised Results section now includes the following:

Multiple regression with executive function being the dependent variable was performed to examine the effect of six-to-eight hour sleep duration compared to other reported sleep times, reflecting a quadratic relationship (**Table 1**). The regression model controlled for age, sleep characteristics including chronotype and report of insomnia, obstructive sleep apnoea traits (daytime sleepiness and snoring) as well as vascular co-morbidity, smoking, BMI, APOE ϵ 4 genotype and socioeconomic status.

3) L172 - Typo - "forty six"

Thank you, this has been amended.

4) L166-196, 190-197 - Were the same sociodemographic and lifestyle covariates used in the Executive Function analyses used here? For consistency and comparability with the cognition measures, these should be added to the model too.

We thank the reviewer for this comment and agree that the same covariates should be considered in the analysis of brain volume. We have created a summary volume measure of all brain regions with a significant quadratic relationship with sleep. This summary measure of sleep-related brain regions was used as the dependent variable in a regression model with the same covariates as in the cognition analysis (sleep characteristics, obstructive sleep apnoea traits, cardiovascular comorbidity, smoking pack years, BMI, genetic risk and socioeconomic status). Importantly, sleeping between six-to-eight hours remained predictive for higher brain volume in these regions.

The revised Methods section now includes the following:

A summary volume measure was calculated for brain regions with a significant quadratic relationship with sleep, described as "sleep-related brain regions", by performing a z-score transformation of each region volume and taking the average across regions. This was used as the dependent variable in a regression model to account for other covariates described above (sleep characteristics, obstructive sleep apnoea traits, cardiovascular comorbidity, smoking pack years, BMI, genetic risk and socioeconomic status).

The revised Results now includes the following:

Using the summary volume measure of these 46 brain regions, sleeping between six-to-eight hours remained predictive of higher brain volume in a regression model which included baseline characteristics of sleep, obstructive sleep apnoea traits, smoking, vascular co-morbidity, APOE ϵ status, body mass index and socio-economic status (Supplementary Table 3).

5) The authors have independently shown that sleep affects brain structure and that sleep affects cognition. Have you considered trying to look at potential causal mechanisms, e.g. that short/long sleep is associated with greater atrophy in certain brain regions which mediate greater impairments in cognition? Or perhaps show overlapping brain regions associated with both sleep-related atrophy + greater cognitive impairments?

The reviewer offers some very pertinent suggestions which we have now implemented. While the cross-sectional nature of our analysis cannot identify causal mechanisms, these are interesting questions and we have performed three additional analyses that may indicate a potential causal direction.

Firstly, we created a mediation analysis model and showed that brain volume partially mediates the relationship between sleep duration and executive function. This was performed using the summary volume measure of 46 brain regions which had a significant quadratic relationship with sleep.

Secondly, we investigated the overlap between brain regions associated with sleep duration and brain regions associated with executive function. Brain volume in 41 regions showed a significant relationship with both sleep duration and executive function. Thirdly, larger brain volumes related to six-to-eight hour sleep duration significantly correlated with the beta-coefficients of brain regions predicting executive function ($r^2 = 0.32$, $*p < 0.001$). These analyses suggest involvement of a widespread sleep-related structural network and offer a putative mechanism linking sleep duration to executive function.

The revised Methods section now includes the following:

Mediation analyses were conducted to estimate the directional influence of brain volume to executive function. Mediation analysis was run in Lavaan, with nonparametric bootstrapping with 1,000 iterations to estimate direct and indirect effects between variables. Linear analyses identified significant associations between individual brain region volume and executive function. These results were corrected for multiple comparisons using the Bonferroni method and these regions were compared with those associated with sleep duration.

The revised Results now includes the following:

Relationship between sleep and executive function is partially mediated by brain volume

Mediation analyses was used to further examine the relationship between sleep duration, brain volume and executive function (Fig. 4d). Sleeping between six-to-eight hours was significantly associated with higher executive function (direct path c) and this relationship was partially mediated by brain volume, using the summary volume measure of sleep-related regions, whereby the relationship between sleep duration and executive function reduced with the inclusion of an indirect path between sleep duration and brain volume (path a) and brain volume to executive function (path b). This reduction was represented by a modest, but significant, drop in beta value of $a*b = 0.01$ ($p < 0.001$).

Brain areas associated with six-to-eight hour sleep duration were also the ones that best predict Executive Function

A comparison of individual brain regions associated with both sleep duration and executive function was performed. Volume in 56 brain regions significantly predicted executive function (Supplementary Table 4) while volume in 46 brain regions showed a significant quadratic

relationship with sleep duration, as shown above. Forty-one of these brain regions overlapped (Supplementary Fig 3) including hippocampi, thalami, orbitofrontal cortex, right frontal pole, left parahippocampal gyri and cerebellar regions. Further, regional brain volume differences (%) between individuals who slept between six-to-eight hours compared to other durations were correlated ($r^2 = 0.32$, $*p < 0.001$) with the strength (beta-coefficients) of regional brain volume predicting executive function (Fig 4e). This showed that regions which were larger in individuals who slept six-to-eight hours were also larger in those with high executive function. Conversely, regions which did not relate to sleep in this way, also did not predict executive function.

We thank the reviewer for their very helpful suggestions.

The revised Figure 2 now includes:

d Mediation analysis, values are standardised beta estimates. Path a: relationship between sleep duration of six-to-eight hours and sleep-related brain region volume. Path b: relationship between sleep-related brain region volume and Executive Function. Path c: direct relationship between sleep duration (6-8 hours) and Executive Function ($a*b$ value is the beta estimate for indirect pathway whereby brain volume mediates the relationship between sleep and Executive Function). All paths were significant at $p < 0.001$. **e** Correlation analysis showing that higher regional brain volume in participants with six-to-eight hour sleep duration correlated with the strength (beta-coefficients) of regional brain volume predicting Executive Function ($r^2 = 0.32$, $*p < 0.001$). Each datapoint is a brain region (example arrow indicating right Accumbens).

The revised Discussion now includes the following:

While a close association between sleep, cognition and brain structure may appear intuitive, this has not been previously been shown in a single cohort, and was supported by mediation analysis findings and the large overlap of individual brain regions associated with sleep and cognition. While causal pathways cannot strictly be inferred, our findings highlight the important relationship between sleep duration, cognition and a large structural network of brain regions.

A new Supplementary Figure 3 has been added:

Supplementary Figure 3. Regional brain volumes which are significantly associated with a higher executive function score and sleep duration six-to-eight hours. Volume in 56 brain regions significantly predicted executive function while volume in 46 brain regions was higher with a six-to-eight sleep duration. Forty-one of these brain regions overlapped.

REVIEWERS' COMMENTS:

Reviewer #1 (Remarks to the Author):

Thanks for the responses. The paper looks like a really useful addition to the literature.

Reviewer #2 (Remarks to the Author):

The authors have satisfactorily responded to my comments.

We would like to extend our thanks to the editor and reviewers for their continued effort and time spent to assess our manuscript. We are grateful to reviewer #1 and #2 for their comments to our revisions.

REVIEWERS' COMMENTS:

Reviewer #1 (Remarks to the Author):

Thanks for the responses. The paper looks like a really useful addition to the literature.

We are grateful to reviewer 1 for their further efforts examining our responses and encouraging statement of our work.

Reviewer #2 (Remarks to the Author):

The authors have satisfactorily responded to my comments.

We thank reviewer 2 for their response and the time taken to review our work.